mechanical engineering

orthogonal decomposition, thermoelastic effect, thermoelastic stress analysis, crack, microbolometer, photovoltaic effect detector

**Author for correspondence:**
C. A. Middleton
e-mail: ceri.middleton@liverpool.ac.uk

# Detection and tracking of cracks based on thermoelastic stress analysis

C. A. Middleton[1], M. Weihrauch[1], W. J. R. Christian[1], R. J. Greene[2] and E. A. Patterson[1]

[1]School of Engineering, University of Liverpool, The Quadrangle, Brownlow Hill, Liverpool L69 3GH, UK
[2]Strain Solutions Ltd, Dunston Innovation Centre, Dunston Road, Chesterfield, Derbyshire S41 8NG, UK

 CAM, 0000-0001-9488-9717; MW, 0000-0002-7352-293X; WJRC, 0000-0003-3638-7297; RJG, 0000-0002-5373-0598; EAP, 0000-0003-4397-2160

Thermoelastic stress analysis using arrays of small, low-cost detectors has the potential to be used in structural health monitoring. However, evaluation of the collected data is challenging using traditional methods, due to the lower resolution of these sensors, and the complex loading conditions experienced. An alternative method has been developed, using image decomposition to generate feature vectors which characterize the uncalibrated map of the magnitude of the thermoelastic effect. Thermal data have been collected using a state-of-the-art photovoltaic effect detector and lower cost, lower thermal resolution microbolometer detectors, during crack propagation induced by both constant amplitude and frequency loading, and by idealized flight cycles. The Euclidean distance calculated between the feature vectors of the initial and current state can be used to indicate the presence of damage. Cracks of the order of 1 mm in length can be detected and tracked, with an increase in the rate of change of the Euclidean distance indicating the onset of critical crack propagation. The differential feature vector method therefore represents a substantial advance in technology for monitoring the initiation and propagation of cracks in structures, both in structural testing and in-service using low-cost sensors.

## 1. Introduction

The first documentation of fatigue failure in metals is often credited to Wilhelm Albert who, in 1838, reported failures due to small repeated loads substantially less than the ultimate tensile strength of the components [1]. Nearly 100 years later, Alan Griffith explained the mechanism of crack formation in

terms of the energy required to break atomic bonds and form new surfaces [2]. However, a practical understanding of metal fatigue in engineering structures was only gained after the Comet aircraft disasters in the 1950s [2]. Fatigue remains the most common cause of failure in the aerospace and automotive industries; and the costs of fracture events have been estimated at approximately 5% of GDP in advanced economies [3]. Hence, it is common practice to inspect engineering structures for evidence of the initiation and propagation of fatigue cracks so that they can be monitored, and appropriate remedial action taken to prevent catastrophic failure. Traditionally, inspections take place during maintenance periods, whose frequency is often dependent on the predicted rate of crack initiation and propagation, i.e. inspections are performed sufficiently frequently that, between inspections, a crack is unlikely to propagate to a critical length at which structural failure occurs. A wide range of non-destructive evaluation (NDE) techniques have been developed to aid inspections [4]. In-service monitoring, or structural health monitoring, can also be used to reduce the downtime required for maintenance inspections and to provide continuous information about the condition of a structure [5]. Structural health monitoring is usually performed with small sensors located in critical regions of a structure and, while networks of sensors can provide coverage of a substantial area, the resolution of the data is inevitably limited. Recently, it has been shown that the measurement of strain fields using optical techniques, such as digital image correlation [6], digital shearography [7] and thermoelastic stress analysis (TSA) [8], can be used to reliably identify damage in composites [9] and cracks in metals [10]. While strain-based damage detection offers the potential significant advantages of universal coverage of a structure and of output in the form of strain fields, from which remnant life predictions can be computed, it has major disadvantages including: the cost, mass and volume of the sensors; the massive quantity of data that is generated and must be processed; and the requirement for some kind of surface preparation, for instance the application of a speckle pattern for digital image correlation. The results from the study reported below demonstrate that TSA can be deployed on a painted surface, using a microbolometer detector for a fraction of the cost of commercially available TSA systems and, by decomposing the data, cracks can be detected and tracked in close to real-time at a resolution at least as good as any other technique.

## 2. Background

In safety critical structures such as airframes, the early detection of damage, and higher resolution information on the behaviour of that damage, would allow earlier or more targeted remedial measures to be undertaken. These measures could lead to an increase in safety, a reduction to environmental impact and economic advantages.

TSA is a full-field technique whose use to detect damage in composites has been previously investigated [11–14]; however, its application to the detection of fatigue cracks in metals is relatively unexplored. TSA exploits the thermoelastic principle to determine surface stresses in a dynamically loaded body. There exists a coupling between the mechanical deformation and the change in temperature of an elastic material; i.e. as a material expands, its temperature decreases (and vice versa). Therefore, as a body is loaded dynamically under adiabatic conditions, the local change in temperature is directly related to the local change in the sum of the principal stresses. Elastic deformation in a structural material typically produces changes in temperature of the order of milli-Kelvin, which can be detected by suitably sensitive infrared sensors [8,15]. Measurements are usually made of the amplitude of temperature resulting from cyclic loading that induces elastic strains and stresses under adiabatic conditions, i.e. at a sufficiently high frequency that there is no significant heat transfer. In these circumstances, the temperature change, $\Delta T$ is given by [8]:

$$\Delta T = \frac{-\alpha T}{\rho c_p} \Delta(\sigma_1 + \sigma_2) = AS, \tag{2.1}$$

where $\alpha$ is the coefficient of thermal expansion, $\rho$ is the density and $c_p$ is the specific heat capacity at constant pressure of the material, and $\sigma_1$, $\sigma_2$ are the principal stresses. While $A$ is a calibration constant and $S$ is the signal recorded using an appropriate infrared detector.

One advantage of TSA over other full-field methods is that it only requires simple surface preparation: a matt surface to produce a uniform high emissivity and low reflectivity (to provide a good infrared signal and avoid motion artefacts respectively). It has been commonplace to use staring-array or focal-plane array photovoltaic effect detectors for TSA measurements. These detectors, which need to be cooled, count infrared photons to provide a measure of the surface temperature of an

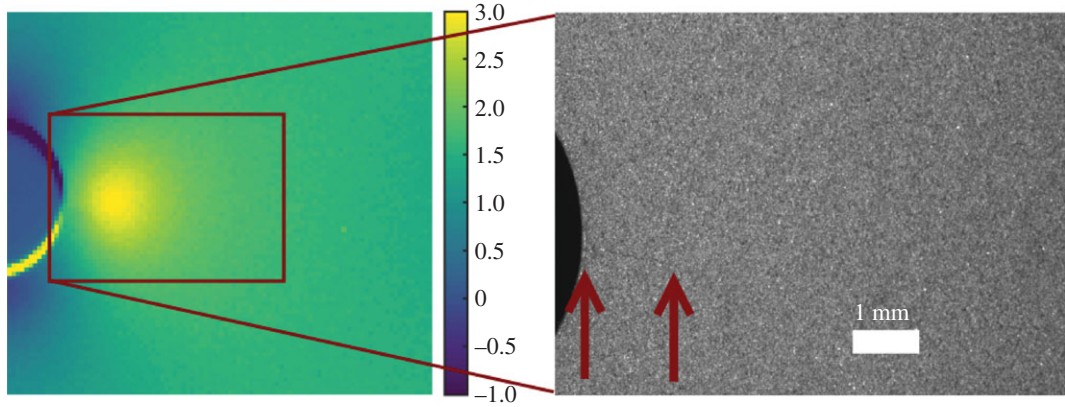

**Figure 1.** Uncalibrated data (normalized using the mean of the field of view) from TSA (left) of a hole in plate specimen after initiation of a fatigue crack, illustrating the TSA patterns indicative of the presence of a crack tip in which the colours are proportional to the amplitude of the temperature $\Delta T$ in equation (2.1); and large magnification image (right) taken with a visual spectrum camera showing the same crack.

object of interest. However, recently the use of microbolometers has been explored successfully [16,17]. In a microbolometer detector, the incident infrared radiation causes heating of the sensor which induces a change in its electrical resistance, and the change in resistance can be related to the change in temperature. These technological developments have made TSA systems orders of magnitude cheaper than those previously available, which, combined with the size difference between photovoltaic effect detectors and microbolometer detectors, means that applications in industrial environments are becoming more feasible [18].

The use of TSA for determining the stress intensity factor associated with cracks, by fitting equations describing the stress field around the crack tip to the measured TSA data, is well established [19,20]. The coordinates of the crack tip can be treated as unknowns in some of these methods and hence the crack tip can be accurately located. Alternatively, the map of the phase difference between the loading signal and temperature response can be used to identify the location of the crack tip, as a point of inflexion between the non-zero phase differences associated with the crack tip plastic zone and the crack wake [21]. Non-zero phase differences occur when adiabatic conditions do not prevail, for instance, due to energy released by dislocation movement in the crack tip plastic zone and frictional heating in the crack wake [22]. However, these techniques assume that the crack tip stress field can be accurately described by linear elastic fracture mechanics and are computationally intensive, require appropriate seeding and high-resolution data obtained using constant amplitude and frequency loading. These drawbacks render these techniques of limited value when using low-cost sensors, with relatively low spatial, temporal and temperature resolution, or in industrial applications where the loading might be of variable amplitude and frequency, such as during flight cycles.

Ancona *et al*. [23] have used TSA to track the propagation of fatigue cracks in martensitic and austenitic steels assuming a prior knowledge of the initial crack. While in 2017, Rajic & Brooks [24] combined a TSA system and an X-Y movement stage to track the propagation of cracks in real time. More recently, Hwang *et al*. [25] have combined passive thermography to identify cracks and active thermography to track fatigue cracks with an accuracy of 99.43% compared with microscopy; however, this system requires a scanning laser which makes its industrial deployment complicated due to safety considerations. Our previous work has introduced the concept of tracking the initiation and propagation of cracks (see for example figure 1) with a method based on the concept of optical flow [10]. This method works well with spatially and thermally high-resolution TSA data, such as collected with a photovoltaic effect detector, and under controlled conditions. However, since the optical flow method is based on identifying apparent movement in a series of images, it is susceptible to errors when applied to data collected using lower resolution sensors or flight cycle loading with varying amplitude and frequency, because the differences between consecutive datasets are too large relative to the differences in the stress magnitudes caused by crack initiation and propagation.

Here, an alternative post-processing approach is considered, which compares the overall shape of the stress distribution in a component over time, using feature vectors obtained through orthogonal decomposition of the TSA data fields. Orthogonal decomposition is a technique that reduces the dimensionality of a two-dimensional matrix of data by representing it using a set of orthogonal

polynomials and assigning the coefficients of the polynomials, sometimes termed shape descriptors, to a one-dimensional feature vector. The technique has been used widely for image analysis (e.g. Hu [26]) with applications such as target recognition [27] and finger print recognition [28]. Fields of displacement or strains can be treated as images and decomposed using, for example, Chebyshev [29] or Zernike [30] polynomials to generate a one-dimensional feature vector which can be readily compared with vectors representing other fields of data.

This process has been implemented in a purpose-written Matlab code that was originally developed as part of an inter-laboratory study performed during the preparation of a CEN guide [31], and developed further into the freely available Euclid software [32]. This technique allows images that originally consisted of $10^4$ to $10^6$ pixels to be accurately represented using typically fewer than $10^2$ coefficients. The list of coefficients for each image are collated into a vector known as a feature vector, which can be used to make computationally efficient comparisons between images. Wang *et al.* [33] showed that strain fields could be decomposed so that measured fields could be used to update finite-element models; while Sebastian *et al.* [34] treated measured and predicted strain fields as images which were decomposed using Chebyshev polynomials to create a pair of feature vectors that could be quantitatively compared as part of a validation process; while Patki & Patterson [35] compared feature vectors based on Zernike polynomials and representing strain fields measured in virgin and impact-damaged composite specimens, using digital image correlation, to quantify the extent of damage; and subsequently Christian *et al.* [30] extended the concept to allow the prediction of residual strength of impacted composite laminates. The change in the shape of the strain field due to the damage was characterized using the Euclidean distance between the feature vectors representing the strain fields in the virgin and damaged states.

In this study, we have applied this decomposition technique to a different type of full-field data, i.e. uncalibrated TSA data which, in effect, are maps of the amplitude of the temperature variation caused by the thermoelastic effect. Whereas Christian *et al.* [30] were able to compare the differences between initial (undamaged) and final (impact damaged) states, the fatigue tests in the current study allow the evolution of the condition of a specimen to be observed from a zero (or low) damage state through to failure. This permits the initiation of damage in a virgin specimen to be detected, as well as the evolution of damage over time to be tracked. The decomposition technique was carried out initially using high-quality data from a photovoltaic effect detector and a specimen subject to constant amplitude and frequency loading; and then using data from a packaged microbolometer detector that is commercially available and a specimen subject to flight cycle loading with variable amplitude and frequency. The difference in cost of the photovoltaic effect detector system and the microbolometer detector was about an order of magnitude; however, the low-cost system was able to identify and track fatigue cracks effectively without applying any special surface preparation to the specimen. Hence, the methodology should be attractive for practical engineering applications.

# 3. The proposed method

Fatigue cracks were grown in specimens made from aerospace grade aluminium alloy, subjected to tensile loading at both constant amplitude and frequency and at variable amplitude and frequency, based on an idealized flight cycle. The specimens were monitored using two types of infrared detector, in terms of cost and resolution, namely: a photovoltaic effect detector and a commercially available microbolometer detector. The temperature data from the detectors were processed to yield fields of uncalibrated TSA data, equivalent to the amplitude of the temperature variation during loading, and each field of data was decomposed as an image to produce coefficients of the Chebyshev polynomials collated into a feature vector. The change in the feature vector was used to identify the presence of a crack and to track its propagation. The details of this procedure are described below and in the flowchart in figure 2, with the results reproduced in the following section.

## 3.1. Specimens

Specimens were machined from one sheet of aluminium alloy 2024-T3, thickness 1.6 mm, with the loading direction perpendicular to the rolling direction of the sheet. A circular hole, 6 mm in diameter, was introduced to the centre of the specimen to generate an initial stress concentration, so that the natural initiation of cracks occurred during cyclic loading. To achieve a uniform emissivity on the surface of the specimens, they were sprayed with a thin layer of graphite (Graphit 33, Kontakt Chemie, CRC

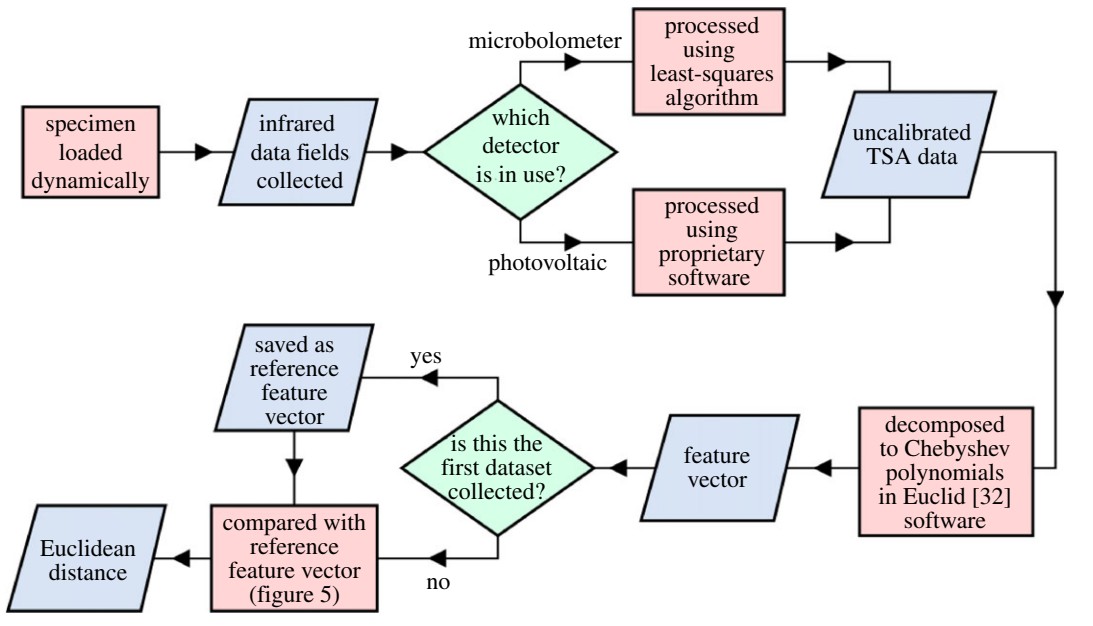

**Figure 2.** Flowchart of the proposed method. Parallelograms represent data (inputs and outputs), rectangular and diamond boxes represent processes and decisions, respectively.

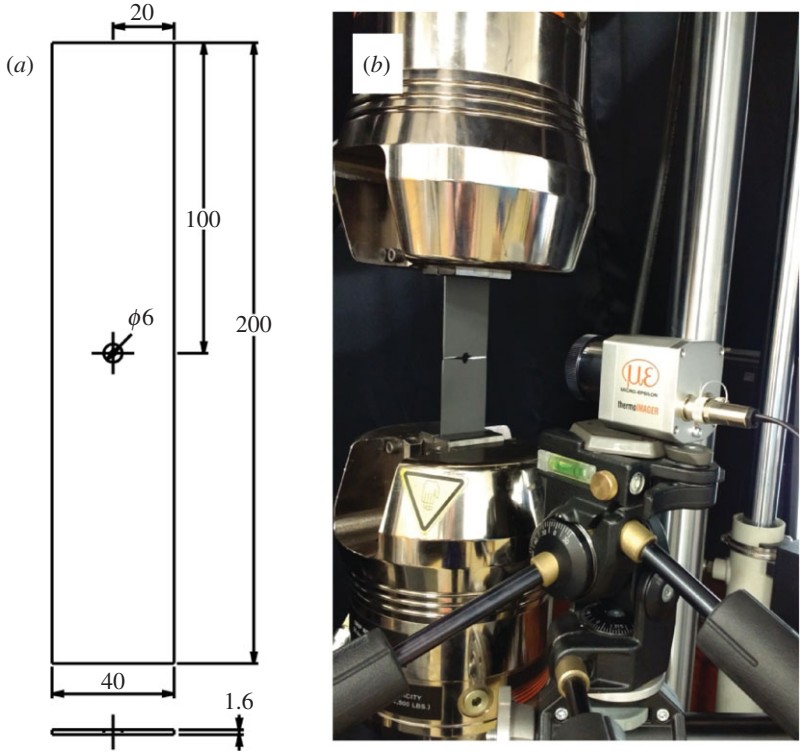

**Figure 3.** (*a*) Specimen geometry (dimensions in mm), and (*b*) photograph of experimental set-up showing the loading machine and the packaged microbolometer detector.

Industries Deutschland GmbH, Iffezheim, Germany); however, one specimen was painted with matt black paint (Plasti-kote 23101 Premium Spray Paint, matt black) as in previous work [10] to allow direct comparison with other tracking methods. The dimensions of the specimen are shown in figure 3.

## 3.2. Loading

The specimens were loaded in an Instron 8501 universal testing machine (Instron, Buckinghamshire, UK), as shown in figure 3, with two loading regimes: (i) constant amplitude and frequency tensile–tensile

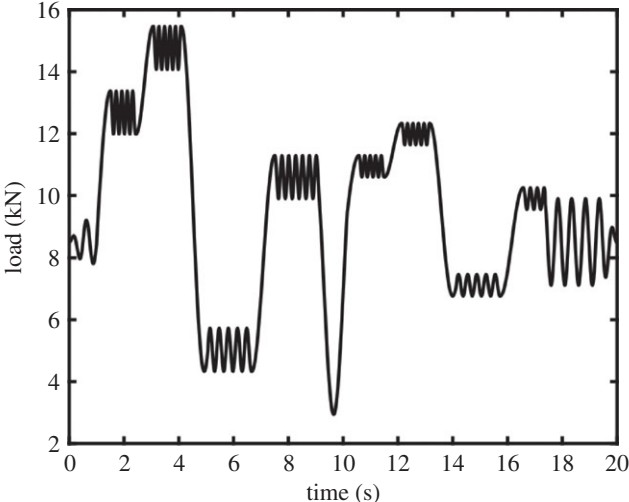

**Figure 4.** An idealized flight cycle with periods of different amplitude, mean and frequencies representing different events within a typical aircraft flight. This flight cycle was repeated continuously so that the specimen experienced multiple transient loading cycles until failure of the specimen.

sinusoidal loading, with $F_{max} = 9.6$ kN, $F_{min} = 0.96$ kN ($R = F_{min}/F_{max} = 0.1$), and (ii) flight cycle loading, with an idealized tensile load cycle as shown in figure 4, which is a simplified representation of the loads experienced by an aircraft component during different events within a typical flight. The frequency of loading during constant amplitude testing was 19 Hz during data collection with the photovoltaic sensor, 13 Hz with the microbolometer.

Loading under flight cycle conditions did not achieve crack initiation on a feasible laboratory timescale. Therefore, specimens were first pre-cracked at constant amplitude conditions ($F_{max} = 12.78$ kN, $F_{min} = 1.278$ kN, 13 Hz) until inspection of TSA data determined a crack had initiated, which was then propagated using flight cycle load conditions.

## 3.3. Data collection and pre-processing

During loading, thermal data were captured with two TSA systems: (i) a high-resolution, high-sensitivity photovoltaic effect detector consisting of a FLIR SC7650 cooled infrared camera with a $640 \times 512$ pixel array, fitted with a 50 mm infrared lens, with an overall size of $232 \times 165 \times 130$ mm, which was integrated with TSA software into a DeltaTherm 1780 system (Stress Photonics, Madison, WI). During constant amplitude loading, data were initially collected with a frame rate of 328 fps, and an integration time of 8 s, using one-quarter of the sensor, i.e. $320 \times 256$ pixels, giving a resolution of 5.8 px mm$^{-1}$, at intervals of 2000 cycles until a crack was detected, after which data were collected at intervals of 250 cycles. For flight cycle loading, data were integrated over 2 s, and collected at nominal 3 s intervals at a resolution of 3.8 px mm$^{-1}$. And, (ii) a lower thermal resolution, microbolometer detector (TIM 450 Thermoimager, Micro-Epsilon Messtechnik GmbH & Co, Ortenberg, Germany) with a $382 \times 288$ pixel array, with a 15 mm lens, measuring $55 \times 46 \times 67$ mm. For this system, infrared frames were collected at 80 fps and TSA data were integrated over 128 frames, or approximately 1.6 s for both constant and flight cycle loading. The spatial resolution was 6.1 px mm$^{-1}$ for constant amplitude data and 4.1 px mm$^{-1}$ for flight cycle data. Both of the detectors operated in the 8 to 14 μm wavelength range.

The data fields from the photovoltaic effect detector were processed to generate the TSA output signal, $S$ in equation (2.1), using the proprietary software supplied with the DeltaTherm system and employing the self-reference option. This option requires the user to define a rectangle in the field of view at the start of loading, and subsequently the data from this rectangle is used by the system's software to identify the frequency and phase of the applied or passing stress field. Hence, it is preferable to define a rectangle of data in a region of uniform, elastic stress; in this work, the self-reference area was defined on the specimen, at some distance from both the stress concentrations generated around the hole and point of load application. The same process was implemented in the software written to handle the data from the microbolometer, which is described below.

An algorithm was developed and implemented in C++ for use with the microbolometer detector to extract the values for the signal $S$, in equation (2.1), at each point in the field of view. The algorithm used

the least-squares method to correlate a reference signal representative of the loading of the specimen to the response of each pixel. The number of frames that the correlation spanned was predefined by the user and set to 128 frames. The signal-to-noise ratio of the data fields was improved by applying a moving mean over 20 data fields, which had the effect of reducing noise in the spatial domain, without eroding the temporal resolution. Lastly, the resulting amplitude data, $S$, were normalized by the mean value of the data field.

Data were collected for constant amplitude and frequency loading using separate test specimens for the two detector systems. However, data from flight cycle loading were collected simultaneously with the microbolometer detector, which was aligned perpendicular to the specimen, and the photovoltaic effect detector, which was aligned at approximately 30° to the same specimen. The depth of field of the lens used with the photovoltaic effect detector maintained all of the specimen in focus and prior work has used this arrangement to allow simultaneous observation with both visible light and infrared cameras [10].

## 3.4. Data post-processing

For the constant amplitude datasets, where loading was monitored before crack initiation, the full-field normalized TSA data were decomposed using Chebyshev polynomials in the freely available software Euclid [32]. The choice of the order of the polynomials (i.e. how many coefficients were used in the decomposition) is an important consideration—too low, and the quality of the reconstruction will be lower, and indications of damage may be lost. There is also a limit on the maximum number of coefficients which can be used to decompose images, related to the number of pixels in the image. However, it is not necessary, nor advisable, to use this maximum limit, as the main advantage of representing large data fields with an order of magnitude smaller vector would be lost, and calculations would become computationally inefficient [36].

For the data in this study, the order of the polynomials was chosen such that increasing the order did not significantly change the Pearson correlation coefficient between the original data field and its reconstruction from the coefficients of the polynomials. This criterion was achieved when 66 coefficients (corresponding to a polynomial of order 10) were employed for data fields sampled at the beginning and end of data collection during constant amplitude and frequency loading, for both instruments. These coefficients formed the elements of the corresponding feature vector representing the data field, except for the first coefficient, which is a piston term that is indicative of the mean value of the data field and, hence, contains no information about the shape of the data. Therefore, the coefficient for this term was excluded from all feature vectors. The overall process is illustrated in the flowchart in figure 5.

The feature vector, $f_i$ (whose elements correspond to the coefficients of the Chebyshev polynomials and for which typical values are shown in the bar charts in figure 5), for each data field that was obtained from the decomposition process was compared with the feature vector for a reference image, $f_{ref}$, by calculating the Euclidean distance, $D$, between them, such that

$$D = \|f_{ref} - f_i\|, \tag{3.1}$$

where $\|\cdot\|$ indicates the vector norm. The reference image was the first TSA dataset collected, which for the constant amplitude data was before crack initiation; but, for the flight cycle, after crack initiation.

For virgin specimens, once this Euclidean distance increased in a continuous manner from a background level, crack initiation was assumed to have occurred. Inspection of the TSA dataset at that time was used to confirm this occurrence, and a region of interest (ROI), which contained the initiated crack, was defined. In this work, a square between the edge of the hole, where initiation occurred, and the edge of the specimen was defined. Here, data are processed for both the ROI and the whole field of view (FOV) for the full dataset. Interrogation of this ROI, through calculation of a Euclidean distance for the ROI alone, allows tracking of a single crack; while the remainder of the field of view could be monitored for the occurrence of further cracks. For flight cycle data, where specimens were pre-cracked, this ROI was predefined for all datasets.

## 4. Results

The Euclidean distances, $D$, calculated using equation (3.1), were normalized using the maximum value found for each test for presentational purposes. These normalized values are shown as a function of time,

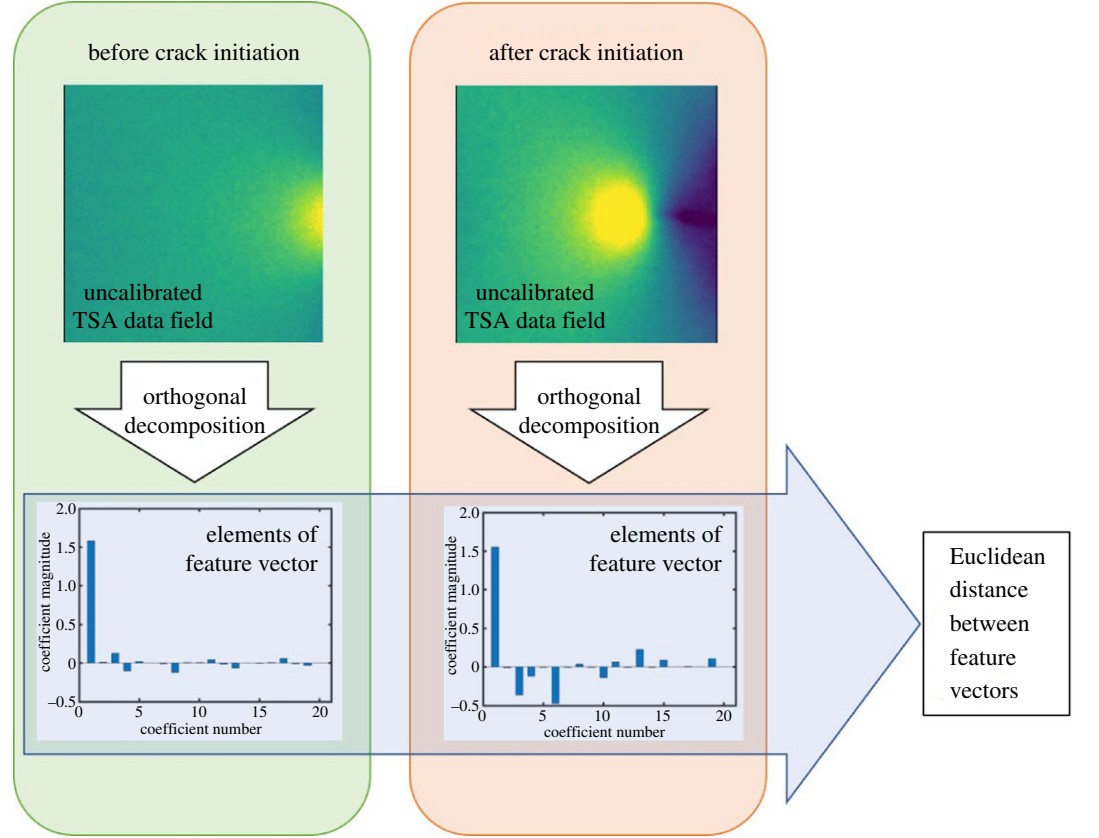

**Figure 5.** Flowchart illustrating the process used in the differential feature vector method, in which an initial TSA data field corresponding to the virgin state (top left) is compared with the current data field (top right) by representing the fields, using an identical decomposition process, as feature vectors (the elements of the feature vectors are plotted in a bar-chart (bottom)) and then the Euclidean distance between the feature vectors is calculated where a non-zero value of the distance indicates a potential change in the condition of the component.

measured in load cycles, for typical constant amplitude and frequency fatigue tests in figures 6 and 7 using the photovoltaic effect and the microbolometer, respectively.

In figure 6, the results are compared with the crack lengths determined (i) using the optical flow method [10], and (ii) from examination of the phase data to identify the position of the crack tip using the method developed by Diaz *et al.* [21]. Individual TSA data fields are also shown at selected cycle numbers, illustrating that changes in the measured optical flow crack length, crack length based on phase information, and the Euclidean distance between feature vectors representing the initial and current state, correspond to qualitative changes in the data fields. The Euclidean distance calculated using both the whole field of view and the defined ROI is approximately zero from 98 000 to 120 000 cycles, i.e. there is no significant change in the shape of the TSA data fields (S in equation (2.1)) and, hence, there is no damage in the specimen sufficient to change the strain field. After 120 000 cycles, there is an increase in Euclidean distance, indicating the initiation of damage sufficient to change the elastic strain field. The Euclidean distance between the feature vectors describing the initial and current state begins to increase gradually and then the rate of increase accelerates until 140 000 cycles, which corresponds to just prior to complete failure of the specimen. The optical flow method indicates crack initiation and propagation slightly later, at 124 000 cycles, than the change in Euclidean distance that occurred at 122 000 cycles.

The equivalent data in figure 7 from the microbolometer detector show similar trends, with the Euclidean distance between the feature vectors representing the initial and current state starting to increase from zero at 55 000 cycles; and then increasing at an accelerating rate towards complete failure. A comparison with the TSA data fields in the inset figures shows that the increase in Euclidean distance corresponds to movement in the stress concentrations associated, initially with the hole, and subsequently with the crack tips. The lower resolution of the data fields from the microbolometer detector was inadequate for the optical flow algorithm and no meaningful results were obtained.

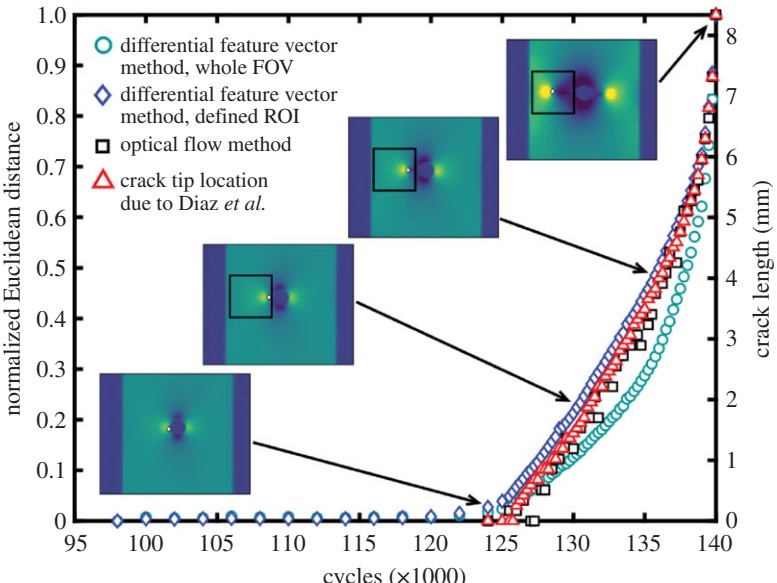

**Figure 6.** Normalized Euclidean distance between the feature vectors representing the initial and current TSA data fields ($S$ in equation (2.1)) collected by the photovoltaic effect detector during constant amplitude and frequency loading as a function of load cycle (blue diamonds; left axis) compared with crack length (right axis) from the optical flow method (black squares) due to Middleton *et al.* [10] and with the crack length calculated from crack tip positions evaluated from the phase difference between the load and temperature signals using the method due to Diaz *et al.* [21]. Insets show the TSA data fields at indicated cycle values. Optical flow data for this specimen were previously presented in [10].

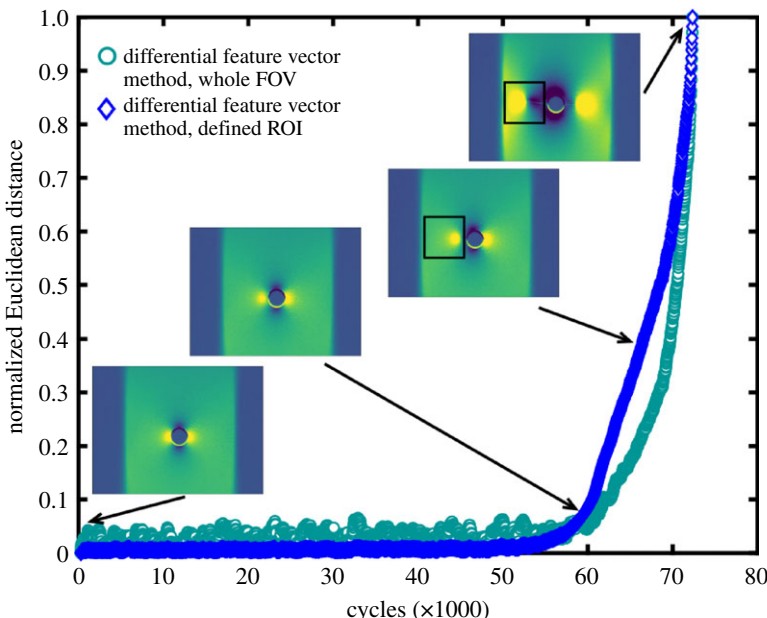

**Figure 7.** Normalized Euclidean distance between the feature vectors representing the initial and current TSA data fields ($S$ in equation (2.1)) collected by the packaged microbolometer detector during constant amplitude and frequency loading, as a function of load cycle. Insets show TSA signal magnitude data at indicated cycle values, and where an ROI was defined for the Euclidean distance calculation, this is indicated on the inset figure.

The corresponding data for typical flight cycle tests on tensile specimens with a central hole are shown in figure 8, based on TSA data fields obtained using the photovoltaic effect and microbolometer detectors, respectively. In these tests, only the differential feature vector method was used, i.e. the data show the Euclidean distance between the feature vectors representing the initial and current data fields of the TSA signal, $S$. These data exhibit substantially more scatter as a consequence of the fluctuations in amplitude and frequency of the loading; and, hence, a moving mean filter (averaging over 20 datasets) was applied to make the overall trend in behaviour more apparent.

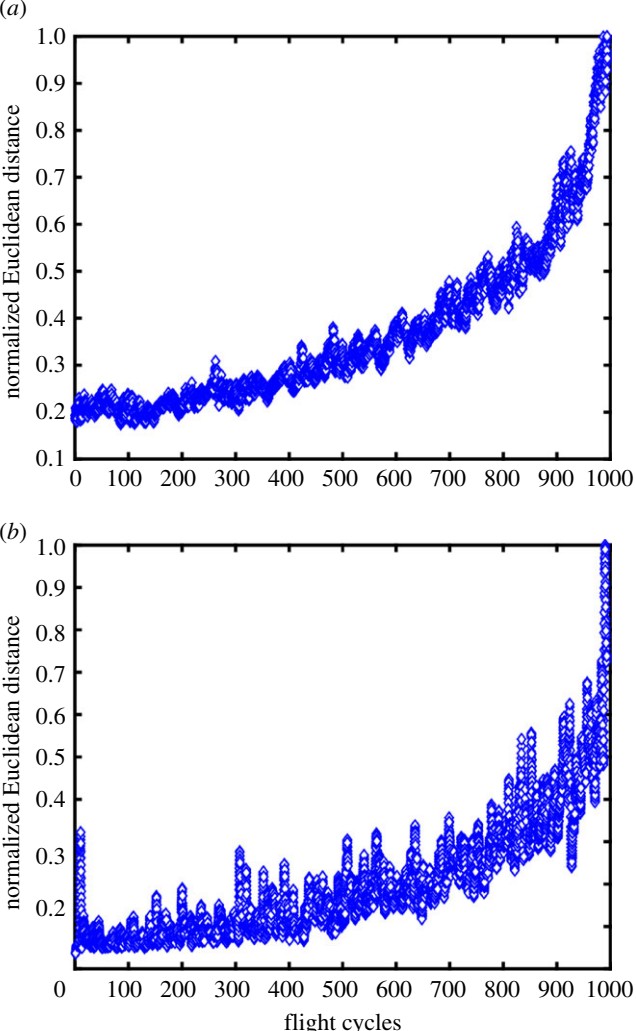

**Figure 8.** Normalized Euclidean distance between the feature vectors representing the initial and current state of TSA data fields ($S$ in equation (2.1)) collected using the photovoltaic effect detector ($a$), and packaged microbolometer detector ($b$) as a function of the number of flight cycles using the cycle shown in figure 4.

## 5. Discussion

The results described in the previous section show that orthogonal decomposition of uncalibrated TSA data can be used to indicate the initiation of damage in the form of cracks during fatigue loading, and as a measure of the propagation of that damage. The TSA data fields shown at selected numbers of fatigue cycles in figures 5 and 6 are qualitatively indicative of a crack initiating and propagating, and the feature vectors obtained from image decomposition give a direct indication of this change when the Euclidean distance between successive vectors increases from approximately zero.

For the specimens in figures 6 and 7, a virgin specimen was subject to fatigue loading, allowing initiation of the crack to be observed. However, a pre-damaged state does not preclude the use of this differential feature vector method. When damage is already present, the reference image is no longer a pristine specimen, rather an initial, partially damaged state, from which Euclidean distances are calculated. When further damage is introduced to the specimen, the Euclidean distance increases as the stress field changes, as seen in the data collected during fatigue loading of pre-cracked specimens in figure 8.

Euclidean distances calculated using the defined ROI following detection of the crack appear to be less noisy than those based on the whole field of view; however, the difference is not significant and does not influence the capability to identify crack initiation. We have shown in previous work [10] that the optical flow method can indicate the initiation of cracks at sub-millimetre lengths; and as can

be seen in figure 6, a change in the Euclidean distance occurs before the equivalent change in crack length indicated by the optical flow method. This suggests that the differential feature vector method, based on Euclidean distance, is more sensitive than the optical flow method, and therefore that it would indicate crack initiation earlier.

The constant amplitude and frequency loading experiments were designed to induce crack initiation and propagation to failure in a relatively short time period; so that the TSA data could be recorded for the entire event in a single session in the laboratory. However, the flight cycle loading was substantially less aggressive, and it was necessary to pre-crack the specimens using constant amplitude and frequency loading to initiate a crack. Hence, it was impractical to demonstrate the initiation of a crack due to flight cycle loading; but, it is anticipated that the sensitivity of the differential feature vector method to crack initiation due to flight cycle loading would be similar to that demonstrated for constant amplitude and frequency loading.

It was found that the optical flow method did not work for the flight cycle data and thus would be unlikely to work for random loading. The optical flow method [10] relies on the only significant change in the stress field between two successive data fields resulting from the initiation or propagation of a crack, which is not the case when the applied loads vary, depending on the relative position of the TSA integration window within the flight cycle. However, because the differential feature vector method exploits the shape of the data and is largely independent of the magnitude of the applied load, it is therefore more robust in a wider range of conditions.

The temporal resolution is limited by the frame rate of the sensor, and the number of frames over which integration occurs. The constant amplitude data were collected over larger time intervals with the photovoltaic effect detector than the equivalent data using the microbolometer; however, it should be noted that this is not a limitation of the photovoltaic effect detector, but rather an experimental decision to enable the comparison with optical flow data. It can be seen in the flight cycle data for the photovoltaic effect detector, and all data for the microbolometer detector, that usable TSA data can be collected over intervals of seconds, demonstrating that damage can be followed at various temporal resolutions.

The photovoltaic effect detector provides high-resolution data in the spatial and temperature domains with the result that the change in Euclidean distance when the crack initiates and propagates is very distinct during constant amplitude and frequency loading. It is particularly encouraging that the differential feature vector method also works well with the smaller, lower thermal resolution and financially accessible microbolometer detector. Original equipment manufacturer (OEM) microbolometers have become available recently at about one-tenth of the cost of a packaged microbolometer used in the work described above. Hence, a further experiment using constant amplitude loading at 1 Hz to propagate a crack was carried out with a low-cost OEM microbolometer detector (Lepton 3, FLIR, Wilsonville, OR), measuring $11.50 \times 12.70 \times 7.14$ mm, which was connected to a dedicated breakout board ($25 \times 30 \times 17$ mm) (Lepton Thermal Camera Breakout v. 1.4, FLIR, Wilsonville, OR) and a low-cost credit card-sized computer (Raspberry Pi 3 B+, Cambridge, UK). This detector captured $160 \times 120$ pixels images at a frame rate of nominally 8 fps using a fixed lens assembly, which gave a spatial resolution of approximately 1.78 px mm$^{-1}$. TSA data were integrated over 180 frames and processed using the same algorithm as for the packaged microbolometer but implemented in Matlab. The results of this additional experiment are shown in figure 9 and demonstrate that the method proposed for detecting cracks and tracking their propagation is sufficiently robust that it can be deployed with a low-cost and low-resolution detector. These results demonstrate that the range of applications for this method are not limited to those for which the large capital cost of a photovoltaic detector can be justified, or where such a detector is already available; nor is it necessary to provide physical access for the large volume and mass associated with the large cooled photovoltaic effect detector. The packaged microbolometer detector is pocket-sized and an order of magnitude less expensive than the photovoltaic effect detector; however, both require power and data cables attached to them which restricts usage in some inaccessible locations in complicated structures. By contrast, the OEM microbolometer is two orders of magnitude cheaper than the photovoltaic effect detector and could be deployed with a battery-powered, low-cost credit card-sized computer, e.g. a Raspberry Pi, for limited time periods. Data transmission could be achieved using, for instance, a Bluetooth module; hence, it offers a flexible low-cost solution for monitoring cracks in structures where power might not be available and data cables might be impractical.

In the data presented here, crack initiation is only indicated by Euclidean distance in the constant amplitude data, where the Euclidean distance increases from approximately zero to a positive number. It is not possible to see an equivalent increase in the flight cycle data, as the initiation stage of the crack development occurred during pre-cracking which was necessary due to time limitations. However, given unlimited time, observation of initiation should be possible, but would probably not

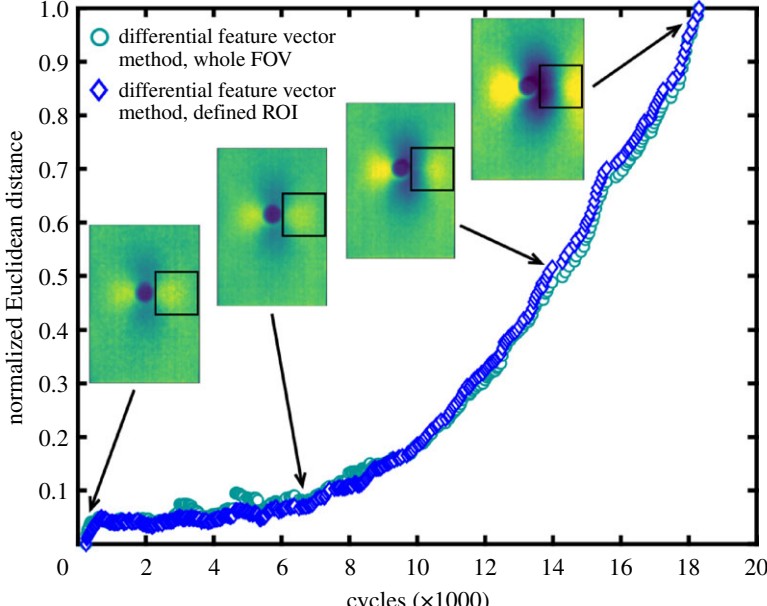

**Figure 9.** Normalized Euclidean distance between the feature vectors representing the initial and current TSA data fields ($S$ in equation (2.1)) collected by the OEM microbolometer detector during constant amplitude and frequency loading, as a function of load cycle. Insets show TSA signal magnitude data at indicated cycle values.

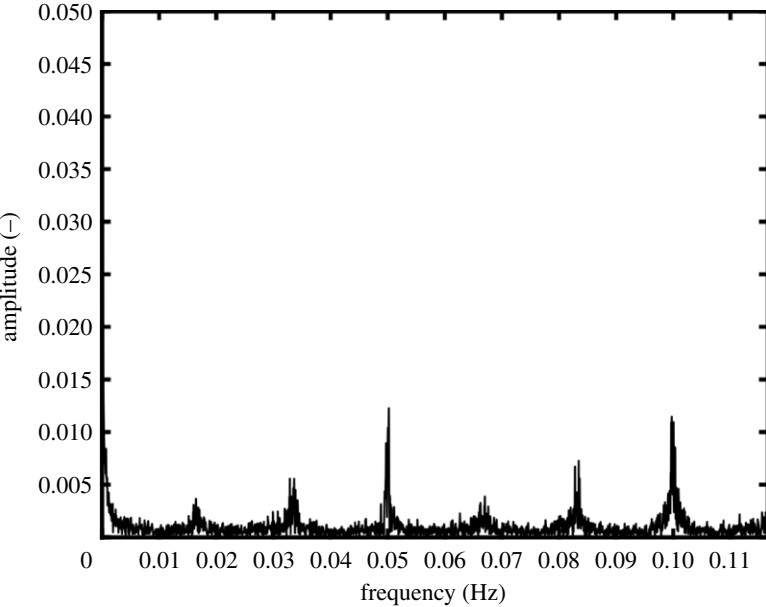

**Figure 10.** Spectrum from a Fast Fourier Transform performed on the Euclidean distance as a function of time, calculated from the data acquired using the photovoltaic effect detector during flight cycle loading (figure 8a). The peak at approximately 0.05 corresponds to the repeat of the flight cycle at 20 s intervals.

be as clear as in the constant amplitude case, due to the oscillations in the Euclidean distance caused by the flight cycle.

The proposed method offers some advantages compared with traditional approaches to TSA. For instance, the temporal variation in Euclidean distance can be treated as a signal and used to extract information about the local loading by performing a fast Fourier transform (FFT). Peaks in the resulting FFT spectrum indicate the frequencies present in the strain field experienced by the specimen, as shown in figure 10 where peaks occur at approximately 0.05 and 0.1 Hz, which correspond to those found in the FFT spectrum of 1000 repetitions of the flight cycle shown in figure 4. Other smaller peaks occur in figure 10 at intervals of 0.0166 Hz and could be artefacts of the

data acquisition frequency. Data would need to be processed at higher rates in order to allow the higher frequency components of the flight cycle to be identified in the FFT spectrum. Since the frame rates of the sensors are 328 and 80 fps, respectively, for the photovoltaic effect detector and microbolometer, the key limitation would appear to be the integration time for the calculation of the TSA output which would need to be reduced with a likely consequential loss of data quality. This information could be useful in full-scale testing environments, where the loads applied to the test article are usually known, but often the local stress history in an area of interest is unknown.

A further advantage of the proposed method is that any non-uniformities in the surface preparation will remain constant over time, and therefore be cancelled out, as only changes in the field will be identified. This is contrary to traditional TSA, where it is necessary to have a surface with a uniform, high emissivity and a low reflectivity; although previous work has shown that TSA can be carried out on specimens coated in aircraft primer paint [10,37].

One disadvantage of the differential feature vector method compared with the optical flow method is that it does not indicate the position of the crack tip, so does not directly determine the location or length of a crack, rather it gives the relative progression of damage. However, the location of a crack and a quantitative measure of crack length could be important in some industrial contexts; and hence, this information could be provided by dividing the ROI into tiles. The TSA data from each tile could be decomposed and the difference feature vector method applied on a tile-by-tile basis. When one tile indicated a change before others, it would show the position of crack initiation and indications appearing in successive tiles would provide information on crack length at the resolution of the tiles. This approach has been used in applying image decomposition to the validation of computational models of structural mechanics to indicate zones in a model where its predictions do not correlate with measurements [38]. A limitation of using tiles to survey an ROI is the spatial resolution of the detectors, which would limit the number of tiles and, thus, the resolution at which cracks could be located and their length monitored. However, the low cost and small size of an OEM microbolometer detector would enable an array of detectors to be deployed; thus, allowing data to be acquired from a large number of relatively small tiles.

The results from this study have demonstrated that TSA can be used for monitoring the initiation and propagation of cracks in laboratory specimens subject to service loads. Its implementation in an industrial environment on real engineering components is the next step. There are certain challenges and limitations to be addressed when transferring this technique from the laboratory to industrial engineering applications, including, for example, the impact of the environment on the technology used. The efficacy of TSA depends on the sensor used, its temporal and thermal resolution, the frequency and amplitude of the loading, and the material observed. Therefore, the appropriateness of this proposed method would be application dependent. In addition, motion and vibration during a full-scale test under multi-axial loading would be larger than in the laboratory tests shown in this study. To avoid false indications of damage due to relative motion between the infrared sensor and the field of view, relative motion could be minimized by mounting the sensors directly on the test article, rather than the tripod-mounted set-up shown here.

For industrial applications, the process could be automated, for example to alert an operator when the Euclidean distance between the feature vector representing the initial, virgin or reference state and the current state deviates significantly from zero, indicating the initiation or growth of damage, or when the rate of increase of the Euclidean distance with time reaches a critical value, indicating incipient failure due to a critical crack propagation rate. This information would allow more effective use to be made of expensive structural tests on large-scale infrastructure, such as airframes. Recent work has shown that monitoring changes in the data fields from TSA is at least as reliable as structural health monitoring based on acoustic emission sensors [39]. Hence, there is also potential to deploy microbolometer detectors for structural health monitoring *in situ*, where non-zero values of the Euclidean distance between successive feature vectors would provide an early indication of damage in the structure. For successful deployment of such a system for structural health monitoring, it would be appropriate to determine the reliability of this technique for detecting cracks under relevant conditions, for example, by determining the probability of detection of damage.

# 6. Conclusion

It has been demonstrated that the initiation and propagation of fatigue cracks in tensile metallic specimens can be reliably tracked using TSA combined with post-processing using orthogonal

decomposition. The orthogonal decomposition was performed using Chebyshev polynomials to generate feature vectors representative of the data fields of signal magnitude obtained from TSA instruments; and the Euclidean distance between feature vectors representing the initial or virgin state and the current state provided an indication of the level of damage in the specimen. Comparisons with an optical method to track damage from the same TSA data fields, and with locations of the crack tip identified from maps of phase difference between the loading signal and temperature response, confirmed the reliability of the differential feature vector method.

The differential feature vector method was found to be sufficiently robust that it could be deployed both with high-cost, high-resolution photovoltaic effect detectors, traditionally used in TSA, and with microbolometer detectors that have lower thermal resolution but are an order of magnitude lower in cost for a packaged microbolometer detector that is commercially available, or two orders of magnitude lower in cost for an OEM microbolometer detector that was combined with a low-cost credit card-sized computer. The robustness of the differential feature vector method also enables the tracking of cracks during flight cycle loading when the changes in strain distribution due to the loading rendered the optical flow method impractical.

These attributes, combined with the ability to capture TSA data from surfaces painted using standard aircraft primer shown previously [10,37], represent a substantial advance in technology for monitoring the initiation and propagation of cracks in structures both in structural testing and in service, with the potential to use arrays of physically small, low-cost detectors for structural health monitoring with automated alerts for the initiation of cracks and information about the presence, location and extent of damage, together with strain field data to support the evaluation of remnant life.

Data accessibility. Data on which this study is based are available from the Dryad Digital Repository at: https://doi.org/10.5061/dryad.s1rn8pk41 [40].

Authors' contributions. C.A.M. and M.W. performed the experimental work and data processing; W.J.R.C. advised on the implementation of image decomposition and calculation of Euclidean distances; R.J.G. and E.A.P. conceived and supervised the work; all authors contributed to the final manuscript following preparation of a first draft by C.A.M. and E.A.P.

Competing interests. The authors have no competing interests.

Funding. The INSTRUCTIVE project was funded by the EU Framework Programme for Research and Innovation Horizon 2020, Clean Sky 2, project no. 686777.

Acknowledgements. We thank two anonymous reviewers for their helpful comments which have improved this manuscript. Helpful discussions and assistance were provided by the topic managers for the INSTRUCTIVE project, Eszter Szigeti (Airbus Operations Ltd) and Linden Harris (Airbus Operations SAS).

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
