## [Reviewer comments · Royal Society Open Science]

Review History

RSOS-200823.R0 (Original submission)

Review form: Reviewer 1

Is the manuscript scientifically sound in its present form?

Yes

Are the interpretations and conclusions justified by the results?

Yes

Is the language acceptable?

Yes

Do you have any ethical concerns with this paper?

No

Have you any concerns about statistical analyses in this paper?

No

Recommendation?

Accept with minor revision (please list in comments)

Comments to the Author(s)

Overall a well written, concise paper in an important topic area.

Just a few comments the authors may want to consider addressing:

1) Parametric Analysis. You mention briefly why you decided to use 66 coefficients. I think it would be helpful to address what happens if you happened to use more or less than the optimum number of coefficients. Would it affect the accuracy of the crack detection?

2) The testing you describe is quite classical (i.e flat specimen with a central notch). You describe/reference some work looking at aircraft testing so I would be interested in knowing how your detection method would react with a much slower test (i.e 0.25 hz vs 1 hz) characteristic of a full scale structural test and either bare aluminum or other low emissivity surface?

3) I would also be concerned that if the loading in the area of interest was complex (multiaxial) with significant movement in the field of view it may degrade the detection methodology. I say this not to take anything away from the current experiment but just to have the authors start thinking about the realities & limitations of using TSA on a full scale test aircraft. Maybe the authors could provide some thoughts about modifications or changes required to apply TSA in an operational setting.

Review form: Reviewer 2

Is the manuscript scientifically sound in its present form?

No

Are the interpretations and conclusions justified by the results?

No

Is the language acceptable?

Yes

Do you have any ethical concerns with this paper?

No

Have you any concerns about statistical analyses in this paper?

No

Recommendation?

Major revision is needed (please make suggestions in comments)

Comments to the Author(s)

Please see the attachment (Appendix A).

Decision letter (RSOS-200823.R0)

Dear Dr Middleton,

The Editors assigned to your paper RSOS-200823 "Detection and tracking of cracks based on thermoelastic stress analysis" have now received comments from reviewers and would like you to revise the paper in accordance with the reviewer comments and any comments from the Editors. Please note this decision does not guarantee eventual acceptance.

Please submit your revised manuscript and required files (see below) no later than 21 days from today's (ie 09-Sep-2020) date. Note: the ScholarOne system will 'lock' if submission of the revision is attempted 21 or more days after the deadline. If you do not think you will be able to meet this deadline please contact the editorial office immediately.

on behalf of Professor R. Kerry Rowe (Subject Editor)
openscience@royalsociety.org

Reviewer comments to Author:
Reviewer: 1
Comments to the Author(s)

Overall a well written, concise paper in an important topic area.

Just a few comments the authors may want to consider addressing:

- 1) Parametric Analysis. You mention briefly why you decided to use 66 coefficients. I think it would be helpful to address what happens if you happened to use more or less than the optimum number of coefficients. Would it affect the accuracy of the crack detection?
- 2) The testing you describe is quite classical (i.e flat specimen with a central notch). You describe/reference some work looking at aircraft testing so I would be interested in knowing how your detection method would react with a much slower test (i.e 0.25 hz vs 1 hz) characteristic of a full scale structural test and either bare aluminum or other low emissivity surface?
- 3) I would also be concerned that if the loading in the area of interest was complex (multiaxial) with significant movement in the field of view it may degrade the detection methodology. I say

this not to take anything away from the current experiment but just to have the authors start thinking about the realities & limitations of using TSA on a full scale test aircraft. Maybe the authors could provide some thoughts about modifications or changes required to apply TSA in an operational setting.

Reviewer: 2

Comments to the Author(s)

Please see the attachment.

===PREPARING YOUR MANUSCRIPT===

===PREPARING YOUR REVISION IN SCHOLARONE===

-- If you have uploaded ESM files, please ensure you follow the guidance at <https://royalsociety.org/journals/authors/author-guidelines/#supplementary-material> to include a suitable title and informative caption. An example of appropriate titling and captioning may be found at https://figshare.com/articles/Table_S2_from_Is_there_a_trade-off_between_peak_performance_and_performance_breadth_across_temperatures_for_aerobic_scorpions_in_teleost_fishes_/3843624.

Author's Response to Decision Letter for (RSOS-200823.R0)

See Appendix B.

RSOS-200823.R1 (Revision)

Review form: Reviewer 1

Is the manuscript scientifically sound in its present form?

Yes

Are the interpretations and conclusions justified by the results?

Yes

Is the language acceptable?

Yes

Do you have any ethical concerns with this paper?

No

Have you any concerns about statistical analyses in this paper?

No

Recommendation?

Accept with minor revision (please list in comments)

Comments to the Author(s)

One final comment regarding your discussion on page 16.

You talk about challenges and limitations of this technology which is being presented as a method for performing structural health monitoring.

In this context, what are your thoughts about the ability of the thermal cameras to make accurate measurements in a flight environment (i.e how would they respond to large temperature differentials, noise and vibration?). Finally, for this technology to be accepted and USEFUL in a flight aircraft you would need to look at determining the POD (probability of detection) of this technology for detecting cracks. This would then allow this TSA based sensor system to be integrated into maintenance & inspection intervals. This is actually a huge hurdle to overcome in the world of SHM and at a minimum it should be acknowledged if not discussed at some level.

Review form: Reviewer 2

Is the manuscript scientifically sound in its present form?

Yes

Are the interpretations and conclusions justified by the results?

Yes

Is the language acceptable?

Yes

Do you have any ethical concerns with this paper?

No

Have you any concerns about statistical analyses in this paper?

No

Recommendation?

Accept as is

Comments to the Author(s)

The authors have adequately addressed my previous comments. I would like to recommend this paper to be published in RSOS.

Decision letter (RSOS-200823.R1)

Dear Dr Middleton

On behalf of the Editors, we are pleased to inform you that your Manuscript RSOS-200823.R1 "Detection and tracking of cracks based on thermoelastic stress analysis" has been accepted for publication in Royal Society Open Science subject to minor revision in accordance with the referees' reports. Please find the referees' comments along with any feedback from the Editors below my signature.

Please submit your revised manuscript and required files (see below) no later than 7 days from today's (ie 16-Nov-2020) date. Note: the ScholarOne system will 'lock' if submission of the revision is attempted 7 or more days after the deadline. If you do not think you will be able to meet this deadline please contact the editorial office immediately.

on behalf of Prof R. Kerry Rowe (Subject Editor)
openscience@royalsociety.org

Associate Editor Comments to Author:

A final recommendation from one of the reviewers to address, but otherwise, your paper is ready for acceptance - congratulations!

Reviewer comments to Author:

Reviewer: 2

Comments to the Author(s)

The authors have adequately addressed my previous comments. I would like to recommend this paper to be published in RSOS.

Reviewer: 1

Comments to the Author(s)

One final comment regarding your discussion on page 16.

You talk about challenges and limitations of this technology which is being presented as a method for performing structural health monitoring.

In this context, what are your thoughts about the ability of the thermal cameras to make accurate measurements in a flight environment (i.e how would they respond to large temperature differentials, noise and vibration?). Finally, for this technology to be accepted and USEFUL in a flight aircraft you would need to look at determining the POD (probability of detection) of this technology for detecting cracks. This would then allow this TSA based sensor system to be integrated into maintenance & inspection intervals. This is actually a huge hurdle to overcome in the world of SHM and at a minimum it should be acknowledged if not discussed at some level.

===PREPARING YOUR MANUSCRIPT===

===PREPARING YOUR REVISION IN SCHOLARONE===

Author's Response to Decision Letter for (RSOS-200823.R1)

See Appendix C.

Decision letter (RSOS-200823.R2)

Dear Dr Middleton,

It is a pleasure to accept your manuscript entitled "Detection and tracking of cracks based on thermoelastic stress analysis" in its current form for publication in Royal Society Open Science.

on behalf of Professor R. Kerry Rowe (Subject Editor)
openscience@royalsociety.org

Appendix A

The authors proposed a TSA method where the feature vectors were calculated from the images, and the Euclidian distance of the features of the pristine and damaged state was used to quantify the damage. The reviewer has a couple of following major comments that are requested to be addressed before this paper can be considered for a possible publication in RSOS.

1. Section 2: the authors did a very good job of providing a thorough review of the state-of-the-art literature on this topic. However, the gap areas of the existing literature were not clearly articulated in the context of the novel contributions of this paper. For example, it was not clear how the proposed approach is substantially different than [31-34]. Also, many equations (Eq. 2-7) were cited in the review section, which is not an appropriate format of a technical paper.
2. Section 3: this section has the title “methods” which is not appropriate; it is suggested to change it to “the proposed method” as it contains all sub-steps of the proposed approach. It is also expected to outline a brief flowchart of the method instead of just detailing all the experimental steps and data processing.
3. The proposed method requires that pristine (i.e., undamaged) state should be available. However, such information may not always be available. Please justify the usefulness of the proposed method.
4. Section 3.4: the feature vectors of Eq. 8 were not clearly explained. It is suggested to include a detailed description of the features that are proposed. The authors showed all results in terms of the Euclidian distance without a standalone demonstration of the feature vectors.
5. Sections 4 and 5 were not articulated appropriately. It is suggested to combine them together and discuss the results immediately after when a particular figure is cited.
6. The context of figure 9 is unclear. Just an FFT at the end of the paper does not add any value and gives rise to many open-ended questions. The authors have used image-based features so far; the context of frequency identification in FFTs is unclear.

Appendix B

Manuscript ID: RSOS-200823: “Detection and tracking of cracks based on thermoelastic stress analysis” Middleton et al.

We thank both Reviewers for their helpful comments, and we have responded (in bold) below and revised the manuscript, tracking and highlighting our amendments.

Response to Reviewer 1.

Comments to the Author(s)

Overall a well written, concise paper in an important topic area.

Just a few comments the authors may want to consider addressing:

1) Parametric Analysis. You mention briefly why you decided to use 66 coefficients. I think it would be helpful to address what happens if you happened to use more or less than the optimum number of coefficients. Would it affect the accuracy of the crack detection?

We have added further explanation of the importance of the choice of number of coefficients, and how this affects the quality of reconstruction and therefore damage detection. A compromise is needed between an ever higher number of coefficients, which would require a large amount of processing power, and a lower number - where questions of accuracy can be raised.

From other work we have carried out (not published), we have found that the differential feature vector method used here benefits from the fact that comparisons between data processed in the same way are made – so we are comparing “like with like”. Therefore, even if the reconstruction is not perfect when compared to the original image, the following images will have the same reconstruction errors, so any difference will be due to damage growth.

2) The testing you describe is quite classical (i.e flat specimen with a central notch). You describe/reference some work looking at aircraft testing so I would be interested in knowing how your detection method would react with a much slower test (i.e 0.25 hz vs 1 hz) characteristic of a full scale structural test and either bare aluminum or other low emissivity surface?

We agree that there are certain limitations on this technique, and it will not be appropriate for all applications. For lower frequency tests, there will be a frequency (sensor, material, and test dependent) under which no further useful TSA data can be collected due to loss of adiabaticity, as mentioned in Section 2. We have added further information clarifying the challenges and limitations of this TSA method to the Discussion.

We have also clarified in Section 2, that for classical TSA, a uniform *high* emissivity and low reflectivity is required, but highlighted the successful use of aerospace primer paint by us (10: Middleton CA, et al. 2019.) and by other authors (37: Rajic N, et al 2014) in the

discussion. It is also possible that our new differential approach presented here allows the surface preparation requirements to be relaxed, as any non-uniformity will be constant through the test.

3) I would also be concerned that if the loading in the area of interest was complex (multiaxial) with significant movement in the field of view it may degrade the detection methodology. I say this not to take anything away from the current experiment but just to have the authors start thinking about the realities & limitations of using TSA on a full scale test aircraft. Maybe the authors could provide some thoughts about modifications or changes required to apply TSA in an operational setting.

We agree that multiaxial loading would complicate the situation using the TSA-derived method here. As the reviewer notes, this was not within the scope of the current work, however we have added some comments on this to the discussion. In particular, relative motion could be minimised by attaching sensors to the test article rather than the external tripod mounting that we have used here.

Response to Reviewer 2.

The authors proposed a TSA method where the feature vectors were calculated from the images, and the Euclidian distance of the features of the pristine and damaged state was used to quantify the damage. The reviewer has a couple of following major comments that are requested to be addressed before this paper can be considered for a possible publication in RSOS.

1. Section 2: the authors did a very good job of providing a thorough review of the state-of-the-art literature on this topic. However, the gap areas of the existing literature were not clearly articulated in the context of the novel contributions of this paper. For example, it was not clear how the proposed approach is substantially different than [31-34].

We have expanded the last paragraph of Section 2 to better explain the literature gap, and clarify that this is the first time this differential feature vector technique has been used on an evolving time series of uncalibrated TSA data.

Also, many equations (Eq. 2-7) were cited in the review section, which is not an appropriate format of a technical paper.

We had thought that the equations provided appropriate background; however, in the light of the reviewer's comment, we have removed these equations, leaving the relevant references.

2. Section 3: this section has the title "methods" which is not appropriate; it is suggested to change it to "the proposed method" as it contains all sub-steps of the proposed approach. It is

also expected to outline a brief flowchart of the method instead of just detailing all the experimental steps and data processing.

We have renamed the section as requested, and included a flowchart of the steps (Figure 2).

3. The proposed method requires that pristine (i.e., undamaged) state should be available. However, such information may not always be available. Please justify the usefulness of the proposed method.

We agree that this was not emphasised sufficiently, but the technique proposed here only requires a change in the condition to occur, not that data from the undamaged state is available.

Initiation of a crack can only be determined from a virgin specimen, but the propagation of a crack in an already damaged specimen can be determined when comparing data from one instant in time to another one at a later instant (as is the case in Figs 8 and 9 where pre-cracking has resulted in damage before the “first” reference image). Therefore, the method is useful where damage may already be present and continuing to grow.

We have edited the relevant parts of the text to clarify that in the first tests here we have started from the undamaged state, but that it can also be applied from an already damaged state. We have changed/edited some terminology e.g. replaced “virgin” (i.e. pristine) with “initial” where appropriate, and emphasised the use of reference data from pre-damaged specimens.

4. Section 3.4: the feature vectors of Eq. 8 were not clearly explained. It is suggested to include a detailed description of the features that are proposed. The authors showed all results in terms of the Euclidian distance without a standalone demonstration of the feature vectors.

We have clarified in the text that a feature vector consists of a list of values of the coefficients of the Chebyshev polynomials (also known as polynomial shape descriptors) generated by the image decomposition, and referenced more clearly the bar charts representing the feature vectors in Figure 5 (previously Figure 4).

5. Sections 4 and 5 were not articulated appropriately. It is suggested to combine them together and discuss the results immediately after when a particular figure is cited.

It was our intention to describe the results in section 4 and then to discuss their overall significance, reliability and potential impact in section 5. We would prefer to make no changes to this structure. We have enhanced the discussion in section 5 in response to the comments from both reviewers and in the process strengthened its narrative thread which we hope goes some way to addressing this comment.

6. The context of figure 9 is unclear. Just an FFT at the end of the paper does not add any value and gives rise to many open-ended questions. The authors have used image-based features so far; the context of frequency identification in FFTs is unclear.

We have added text to better explain the context behind the use of FFT to reconstruct the loading cycle. In this test, we know the flight cycle, but in full-scale testing environments, this will not always be the case. Therefore, if frequency elements of the load cycle can be identified by processing the Euclidean distance, this will give the test operator further information about the loads experienced in the area being monitored.

Manuscript ID: RSOS-200823.R1: “Detection and tracking of cracks based on thermoelastic stress analysis” Middleton et al.

We thank both Reviewers for their second review of our manuscript. We have responded (in bold) below to the final comment and revised the manuscript, tracking and highlighting our amendments.

Response to Reviewer 1.

Comments to the Author(s)

One final comment regarding your discussion on page 16.

You talk about challenges and limitations of this technology which is being presented as a method for performing structural health monitoring. In this context, what are your thoughts about the ability of the thermal cameras to make accurate measurements in a flight environment (i.e how would they respond to large temperature differentials, noise and vibration?).

Finally, for this technology to be accepted and USEFUL in a flight aircraft you would need to look at determining the POD (probability of detection) of this technology for detecting cracks. This would then allow this TSA based sensor system to be integrated into maintenance & inspection intervals. This is actually a huge hurdle to overcome in the world of SHM and at a minimum it should be acknowledged if not discussed at some level.

We agree that the move from laboratory work as shown in this study to implementation in flight aircraft is not a simple one, and would require further testing at a larger range of conditions (both environmental and loading), and consideration of how these sensors respond to those conditions. We have added comments to the discussion to clarify that the impact of the surrounding environment on the sensors, and motion due to vibration must also be considered.

The details of how this technique might be integrated into a full-scale testing system for SHM is outside the scope of this paper, but the problem of how it would be implemented is an interesting one:

- 1) Temperature differential: the difference in temperature between an aircraft on the ground and in the air can be large. Where the absolute temperature has a significant effect on the TSA signal magnitude using traditional TSA, the advantage of this novel technique, is that by decomposing the TSA “images” to feature vectors, the magnitude information is removed, and focus is entirely on the shape of the stress concentrations. So, removing magnitude changes caused by temperature evolution during flight.**
- 2) Noise: noise would have no effect on the IR sensors.**
- 3) Vibration: Problems would occur if vibration is occurring at a comparable frequency to the excitation frequency of the airframe. A solution to this**

would be to damp this vibration, removing relative motion between sensor and locally observed analysis area on the airframe.

We also agree that, before it could be implemented, a determination of the reliability of this technique for detecting cracks in appropriate SHM settings would be necessary. We have added a comment to the discussion to expand on this.

Response to Reviewer 2.

Comments to the Author(s)

The authors have adequately addressed my previous comments. I would like to recommend this paper to be published in RSOS.

We thank the Reviewer for their review, and are pleased that our previous edits have fully addressed all comments.